# Biodegradable PGA/PBAT Blends for 3D Printing: Material Performance and Periodic Minimal Surface Structures

**DOI:** 10.3390/polym13213757

**Published:** 2021-10-30

**Authors:** Zihui Zhang, Fengtai He, Bo Wang, Yiping Zhao, Zhiyong Wei, Hao Zhang, Lin Sang

**Affiliations:** 1School of Automotive Engineering, Dalian University of Technology, Dalian 116024, China; zhangzihui@mail.dlut.edu.cn; 2Department of Radiology, Second Affiliated Hospital of Dalian Medical University, Dalian 116027, China; hft19940214@163.com (F.H.); emmazhaochina@163.com (Y.Z.); 3School of Materials Science and Engineering, Dalian University of Technology, Dalian 116024, China; BobWang@mail.dlut.edu.cn; 4Department of Polymer Science and Engineering, School of Chemical Engineering, Dalian University of Technology, Dalian 116024, China; zywei@dlut.edu.cn; 5Department of Orthopedics, Affiliated Dalian Municipal Central Hospital, Dalian Medical University, Dalian 116027, China

**Keywords:** biodegradable polyesters, polyglycolic acid (PGA), fused deposition modeling (FDM), triply periodic minimal surfaces (TPMS), mechanical property

## Abstract

Biodegradable polymers have been rapidly developed for alleviating excessive consumption of non-degradable plastics. Additive manufacturing is also a green energy-efficiency and environment-protection technique to fabricate complicated structures. Herein, biodegradable polyesters, polyglycolic acid (PGA) and poly (butyleneadipate-co-terephthalate) (PBAT) were blended and developed into feedstock for 3D printing. Under a set of formulations, PGA/PBAT blends exhibited a tailored stiffness-toughness mechanical performance. Then, PGA/PBAT (85/15 in weight ratio) with good thermal stability and mechanical property were extruded into filaments with a uniform wire diameter. Mechanical testing clearly indicated that FDM 3D-printed exhibited comparable tensile, flexural and impact properties with injection-molded samples of PGA/PBAT (85/15). Furthermore, uniform and graded Diamond-Triply Periodic Minimal Surfaces (D-TPMS) structures were designed and successfully manufactured via the fused deposition modeling (FDM) technique. Computer tomography (CT) was employed to confirm the internal three-dimensional structures. The compressive test results showed that PGA/PBAT (85/15) D-surface structures bear better load-carrying capacity than that of neat PGA, giving an advantage of energy absorption. Additionally, typical industrial parts were manufactured with excellent dimension-stability, no-wrapping and fine quality. Collectively, biodegradable PGA/PBAT material with good printability has great potentials in application requiring stiffer structures.

## 1. Introduction

In recent years, fused deposition modeling (FDM) has been developed rapidly among 3D printing techniques because of its low cost in maintenance, and diversity in thermoplastic feedstock [1,2,3]. It enables the production of custom parts with complex structures in many application fields including the medical, food, automotive, aerospace and construction industries [4,5,6]. Among the commercial thermoplastic materials used in FDM 3D printing, poly (lactic acid) (PLA) is rather popular in 3D printing because of its biodegradability, biocompatibility, favorable mechanical properties and facile printability. Nevertheless, PLA-based materials have inherent limitations of brittleness and low toughness [7,8]. With the request of environmental protection and the growing demands in biodegradable polymers, considerable interest has been attracted to develop various biodegradable polymers as feedstock for 3D printing [9,10,11,12].

Polyglycolic acid (PGA), a biodegradable polymer, can be degraded to carbon dioxide and water with a relatively fast degradation rate [13,14,15]. Its degradation products can be absorbed by the human body, which is approved by US Food and Drug Administration (UFDA). Besides, PGA also has superior mechanical strength to other biodegradable polymers [16,17]. It is found that the mechanical strength and modulus are similar to that of human bones, which make it an ideal candidate for hard tissue implanted materials. Wu et al. [18] fabricated polyetheretherketone (PEEK)/polyglycolide acid (PGA) scaffolds using 3D printing technology showed the ability to efficiently sustain drug release as an implant for treating bacterial infection. Taegyun et al. [19] 3D-printed PGA/hydroxyapatite composite scaffolds and demonstrated it can promote patient-specific bone regeneration. The existed literature mainly reported on the incorporation of PGA with other plastics or bioactive fillers, however, few studies focused on the PGA-based filaments for FDM 3D printing and their printability and precision of complex structures.

Currently, biodegradable elastomeric polyesters, such as poly (butyleneadipate-co-terephthalate) (PBAT) [20], poly (butylene succinate) (PBS) [21] and polycaprolactone (PCL) [22] have developed vigorously. It is hoped that blending PGA with these ductile polymers will develop 3D-printing feedstock with good biocompatibility, tailored biodegradability and mechanical properties. Among them, PBAT possesses an excellent ductile property with high elongation at break. Therefore, PBAT is considered a good candidate to improve the flexibility of PGA. It is expected that PGA/PBAT blends could be novel 3D-printing feedstock due to the high mechanical strength of PGA and the elevated toughness of PBAT.

Although there was little evidence of PGA processed by FDM printing in the past, extruding, sintering and injection molding has been attempted [23,24,25]. Additionally, PBAT-based materials (i.e., PBAT/PLA) have also been fabricated into 3D-printing feedstock [26,27]. Therefore, binary PGA/PBAT blends for FDM 3D printing are proposed and subjected to this research. An important aspect of this system is the phase compatibility of the blend, which would directly influence the mechanical performance. Thus, an epoxy-functionalized chain extender was used to react with the carboxyl and carboxyl functional groups in both polyesters [28,29].

The focus of this article is to explore the feasibility of novel PGA/PBAT blends as 3D printing feedstock and manufacturing complex lattice structures with good quality. Firstly, different compositions of PGA/PBAT blends were compounded, the thermal behavior, thermal stability and mechanical performance were subsequently evaluated and the optimization of PGA/PBAT blends for FDM printing was obtained. Then, the emphasis on fabricating PGA/PBAT filaments was carried out, and the mechanical properties of FDM-printed and injection-molded samples were comprehensively evaluated. Finally, periodic minimal surface structures with constant-thickness and graded-thickness were designed and manufactured, and the corresponding compressive performance was compared with the pure PGA group. In summary, this work is to explore the potentials of novel biodegradable PGA/PBAT filament in complex shape, high strength, and lightweight engineering applications in a sustainable and energy conservation way.

## 2. Materials and Methods

### 2.1. Materials

Commercial PBAT having a density of 1.21 g/cm^3^, melting temperature of 125 °C and a melt flow rate (MFR) rate of 44 g/10 min (230 °C and 2.16 kg) was obtained from Kanghui New Materials Hi-Tech Co., Ltd. (Yingkou, China). PGA having a density of 1.64 g/cm^3^, melting temperature of 220 °C and a melt flow rate (MFR) rate of 40 g/10 min (230 °C and 2.16 kg) was kindly provided by Shanghai Pujing Chemical Industry Co., Ltd. (Shanghai, China). Both PGA and PBAT were fully biodegradable polyesters, and the chemical structures were given in Figure 1. A multi-functional epoxy chain extender styrene-glycidyl methacrylate (Joncryl ADR 4370) was purchased from BASF Chemical Company (Ludwigshafen, Germany).

### 2.2. Sample Preparation

Before the extrusion processing, the PBAT pellets were vacuum-dried at 80 °C for 8 h to remove the moisture. PGA pellets were kept in vacuum-sealed bags with a desiccant at 4 °C and dried at 60 °C for 2 h. In order to avoid undesirable hydrolysis during extruding, the predrying process was conducted to remove moisture. Joncryl ADR was used as received. The mass formulations of PGA/PBAT composites contained 100/0, 95/5, 85/15 and 75/15, and the content of Joncryl ADR was 1.5 wt% of the whole biodegradable polyesters. The PGA and PBAT pellets were compounded by a twin-screw extruder (SHJ20-X40, L/D = 40, D = 40 mm, Nanjing Giant Machinery Co., Ltd., Nanjing, China). The processing temperatures of extruding zones were set from 210–230 °C with a rotation speed of 40 rpm.

Test specimens for the tensile test were molded using injection molding equipment (Wuhan Ruiming Machinery Co., Ltd. Wuhan, China) for each blend composition (PGA/PBAT: 100/0, 95/5, 85/15, 75/25). The heating zone of the injection molding system was 230 °C and the molding zone was 40 °C. The standard dumbbell-shaped specimens (ISO 527, type2) were prepared and stored in a sealed dryer before characterization.

### 2.3. Filament Feedstock Fabrication and 3D Printing

Under the optimized formulations of PGA/PBAT composites, PGA/PBAT (85/15) was adopted to fabricate into filaments (1.75 ± 0.5 mm diameter) using a desktop single-screw filament extruder (Wellzoom C, Shenzhen Mistar Technology Co., Ltd., Shenzhen, China). The extruder barrel heating zone to die temperature was set at 225 and 230 °C, respectively. The extruded filaments were pulled through and collected by a winding unit. The filaments were dried in an air-circulating oven at 60 °C for 4 h and stored in sealed vacuumed bags with desiccant (4 °C) prior to 3D printing or other characterization.

3D-printed PGA/PBAT samples were manufactured via an FDM 3D printer (FUNMAT HT, INTAMSYS Co. Ltd., Xi’an, China). The FDM printing condition was set as: a nozzle diameter of 0.40 mm, nozzle temperature at 230 °C, as building platform temperature at 45 °C, ambient temperature at 45 °C, infill density of 100%, printing speed maintained at 20 mm/s, raster angle of 45° and each layer thickness of 0.10 mm.

### 2.4. Design of the Diamond-Triply Periodic Minimal Surfaces (D-TPMS) Structures

Triply periodic minimal surfaces (TPMS) can be mathematically approximated using implicit methods [30,31]. Among types of TPMS structures, Diamond (D) surface was selected in the current work. The D-surface is described as follows:(1)$$\begin{array}{c} \phi D(x,y,z)=\sin(\omega )\sin(\omega y)\sin(\omega z)+\cos(\omega x)\sin(\omega y)\sin(\omega z)+\\ \sin(\omega x)\cos(\omega y)\sin(\omega z)+\sin(\omega x)\sin(\omega y)\cos(\omega z)=C \end{array}$$
where *x*, *y*, *z* represent spatial coordinates, *w* = 2π/l and l is the length of a unit cell. The 3D D-surface is generated as the solution of the level-set function *ϕ* = C. The solid model of Diamond surfaces was created by extracting the zero-level set surface (when C = 0) from Equation (1). Matlab scripting was used to generate the sheet surfaces. The total cylinder sample size has a diameter of 20 mm and a height of 20 mm. Constant-thickness of structures with nominal average wall thickness was 0.4 mm and 2 mm, respectively. The wall thickness of graded structures ranged from 0.4 mm to 2 mm radically. The resultant 3D stereolithography (STL) models were then transferred to CURA software (Ultimaker Co. Ltd., Amsterdam, Holland) for slicing in preparation for 3D printing.

### 2.5. Characterization

Differential scanning calorimeter (DSC). The non-isothermal crystallization and melting behavior of PGA and PGA/PBAT were studied under a nitrogen atmosphere (10 °C/min). The weight of test samples was 5–8 mg and sealed in an aluminum pan. The running program was divided into three stages: heating room temperature to 250 °C, annealing at 250 °C for 5 min; cooling to 20 °C, maintaining for 3 min; reheating to 250 °C.

Thermogravimetric analysis (TGA). Thermal decomposition stability of PGA and PGA/PBAT composites was carried out through TGA (Q600, TA instruments, NewCastle, America). The heating program was from 30 °C to 600 °C under a nitrogen atmosphere at a heating rate of 10 °C/min.

Scanning electron microscopy. The compatibility of PGA and PBAT in injection-molded and 3D-printed samples was observed by a tungsten filament scanning electron microscope (SEM, QUANTA 450, FEI, Hillsboro, America). The fractured surface was spray-covered with a thin gold layer.

### 2.6. Mechanical Test

The mechanical properties of injection-molded and FDM-printed samples were tested by using a mechanical testing machine (GT-7001-HC6, GOTECH TESTING MACHINE INC, Taiwan, China). The tensile strength and modulus were measured at a constant crosshead speed of 2 mm/min at an ambient temperature according to the standard of ISO 527. For flexural tests according to ISO 178 standard, three-point bending with a crosshead speed of 2 mm/min was performed at room temperature.

The stiffness of the FDM-printed D-surface TPMS structures was evaluated from compression tests, according to the ASTM D-695 standard. The unconstrained cellular structure samples were 20 mm in diameter and the height was 20 mm, which were compressed between two rigid flat steel plates with a constant strain rate of 2 mm/min. The compressive force and displacement data from the universal machine were recorded. When the strain of cellular structures reached 20%, the tests were terminated.

The total energy absorption (*EA*) was calculated from the area under the force-displacement curve as follows, Equation (2) [32,33]:(2)$$EA=\int_{a}^{b}\sigma \cdot d\varepsilon$$
where *σ* assigns to the compressive stress and the ε is the nominal strain. The calculation of *σ* = F/A and *ε* = *δ*/H, respectively. F and *δ* corresponded to the compressive force and displacement, which are recorded during the compression test. A is the original cross-section area and H is the height of the D-surface TPMS structure along the compressive direction.

### 2.7. Computed Tomography

The deformation of the D-surface TPMS structure samples after compression was scanned by Computed tomography (CT equipment, SIEMENS SOMATOM DRIVE, langen, Germany). CT scan conditions as follows: the voltage was 70 kV, the current was 61 mA, slice thickness was 0.5 mm, slice spacing was 0.3 mm, FOV (field of view) was 50 mm, the matrix was 512 × 512, and DLP (dose length product) was approximately 45.58 mGy. The obtained scan data were subsequently reconstructed using SIMENS software analysis system (Syngo CT VA62A, Erlangen, Germany).

## 3. Results and Discussion

### 3.1. Preparation and Characterization of PGA/PBAT Samples

The epoxy group of ADR is expectable to react with both hydroxyl and carboxyl of the polyester [34,35]. The introduction of ADR efficiently facilitated the reaction between PGA and PBAT, which resulted in a polymer network (as illustrated in Figure 1) and decreased both the number of hydroxyl and carboxyl end-groups in PGA and PBAT. The cross-linking reaction would increase the melt strength and protect the sensitive groups from hydrolysis degradation during the extruding process.

Prior to filament fabrication and FDM printing, thermal behavior and stability of filaments are essential due to the necessary information on the printed component of the printing window. The DSC thermograms of PGA, PGA/PBAT (95/5), PGA/PBAT (85/15) and PGA/PBAT (75/25) were shown in Figure 2a, b. After eliminating thermal history, the data in the first cooling and second heating scan were collected and plotted into curves. The values of crystallization temperature (T*_c_*), the crystallization enthalpy (ΔH*_c_*) in the first cooling curves; the melting temperature (T*_m_*) and melt enthalpy (ΔH*_m_*) in the second heating curves were summarized in Table 1. It can be seen that T*_c_* of the PBAT (~74 °C) was not detected from the cooling curves [36], while T*_c_* of PGA (~190 °C) was obvious [37]. The crystallization peaks were related to the addition of PBAT, the T*_c_* migrates to higher temperatures from 185.1 to 194.5 °C, which can be assigned to the heterogeneous nucleation effect of the branch chain of PBAT for the crystallization of PGA. However, the ΔH*_c_* was continuously decreased when the content of PBAT reached 25 wt%. This was because that small content of PBAT acted as a dispersed phase while PGA was the continuous phase. The increased content of entangled chains of PBAT might influence the organized PGA polymer crystallization. In the second heating curve, there is only one melting peak at 220.5 °C for neat PGA (100/0), whereas peaks are split into two in PGA/PBAT blends. With the increased content of PBAT, two melting peaks became evident. This was probably due to the formation of two crystalline structures of PGA co-existing in the binary blends.

Furthermore, weight loss and differential thermogravimetric curve (DTG) curves of neat PGA and PGA/PBAT blends were plotted in Figure 2c,d. For neat PGA, the loss weight curve showed that the weight started to decrease at 339.2 °C, and weight loss was quite obvious at 389.7 °C. After incorporation of 5 wt% and 15 wt% PBAT, the initial decomposition temperature slightly increased to 358.2 °C and 346.7 °C, which was probably due to enhanced stability by the cross-linking reaction using ADR. The fastest decomposition temperatures were also improved for PGA/PBAT (95/5) and PGA/PBAT (85/15) (as shown in Table 1). However, when the content of PBAT achieved 25 wt%, the characteristic decomposition peak of PBAT could be detected, and thus the decomposition temperature shifted to a lower temperature at 326.0 °C. Two remarkable decomposition stages were observed at 351.2 °C and 412.5 °C in PGA/PBAT (75/25). The first stage corresponded to the decomposition of PBAT, and the latter peak was assigned to the decomposition of PGA. These results illustrated that the thermal degradation of PGA/PBAT (95/5) and PGA/PBAT (85/15) could be effectively postponed after the extruding process in the presence of ADR, which was suitable as filament feedstock.

The tensile properties of injection-molded (IM) PGA/PBAT with different formulations were presented in Figure 3. The results showed that the tensile strength and Young’s modulus gradually decreased with the increase in PBAT content. The tensile strength of neat PGA was 114 ± 1.24 MPa and Young’s modulus was 5.15 ± 0.15 GPa, which was even stiffer to poly(ether-ether-ketone) (PEEK) materials with a tensile strength of 100 MPa and Young’s modulus of 3.7 GPa [38]. However, excess strength might bring difficulties (i.e., brittleness) in plastic processing and application fields. Since PBAT was an excellent elastomeric “soft” polyester, although its tensile strength was 18 MPa, the modulus was 800 MPa and the elongation at break could reach to ~800%. Therefore, in the current work, PBAT first attempted to modify PGA to broaden the applications. From Figure 3a, the tensile strength gradually decreased from 114 MPa (PGA) to 79 MPa (95/5), 60 MPa (85/15) and 45 MPa (75/25), respectively. On the contrary, the elongation at break was 2.3%, 4.5%, 15.6% and 20.2% for PGA (100/0), PGA/PBAT (95/5), PGA/PBAT (85/15) and PGA/PBAT (75/25), respectively (Figure 3b). Accordingly, although the stiffness of PGA/PBAT was weakened to some extent, elongation at break was significantly increased after incorporation with elastomeric PBAT.

To date, a great variety of synthetic polymers have been developed and used as feedstock materials, such as ABS, PA6, POM and PLA. Among them, PLA was the only commercial biodegradable filament, while other kinds of biodegradable polymer feedstocks were limited. In our work, three formulations of PGA/PBAT blends were fabricated. Among these formulations, the tensile strength and modulus of PGA/PBAT (85/15) were comparable with PLA (tensile strength: 65 MPa, modulus: 2.1 GPa) [39]. Moreover, its ductility was improved for the incorporation of PBAT, which was superior to neat PLA. In previous researches, some elastomeric polymers such PBS, PBAT and other polyesters were also applied to blend with PLA to overcome the inherent brittleness [40]. However, the mechanical properties of binary PLA-based materials were inferior to PGA/PBAT (85/15) prepared in the current work. Accordingly, PGA/PBAT (85/15) was supposed to be a good candidate material as a 3D-printing feedstock with balanced strength and toughness.

### 3.2. PGA/PBAT Filament Feedstock via 3D Printing

Based on the above analysis, neat PGA and PGA/PBAT (85/15) blends were fabricated in 3D printing filaments through a single screw extruder. As shown in Figure 4, uniform and standard filaments with a 1.75 mm diameter were obtained. According to T*_m_* in Table 1, the temperature nozzle at 230 °C could successfully print the PGA/PBAT into the tensile specimens without any defects. The good printability of PGA/PBAT showed great potentials in the additive manufacturing of complicated parts.

Mechanical properties of the 3D-printed PGA/PBAT (85/15) specimens were comprehensively evaluated to compare with injection-molded (IM) control groups (Figure 5, Figure 6 and Figure 7). The tensile, flexural and notched impact tests of the PGA/PBAT (85/15) specimens processed by injection molding and 3D printing were conducted. In Figure 5a, the tensile stress-strain curves of IM and 3D-printed samples showed a similar trend with an obvious necking stage. The elongation at breakage was all above 10%, and the value was higher in IM specimens. Ductile behavior with necking characteristics was observed, suggesting an improved toughness due to PBAT incorporation. In Figure 5b, the tensile strength and Young’s modulus of 3D-printed specimens reached 56.5 MPa and 2.48 GPa respectively, which were equivalent to 94% and 85% of the injection-molded groups, respectively. A slightly lower value in the 3D-printed groups was determined by the layer-by-layer deposition mode. The delamination failure mode between printing layers may obstruct the force transfer when suffering from the external loading force.

Similar results were seen in flexural and impact results (as shown in Figure 6 and Figure 7) Compared with tensile performance, the difference between 3D-printed and IM groups flexural properties was slightly evident. The flexural strength and modulus of 3D-printed specimens were 80% and 82% of those of IM counterparts. This was mainly because the printing direction was vertical from the loading, which brought a challenge for the interface between the printing layers deposition technique. This layer-by-layer limitation was intensified using crystalline or semi-crystalline polymers, which might shrink or delaminate during the printing process. In Table 1, the enthalpy of crystallization was decreased with the increasing addition of PBAT, suggesting an inhibition effect of PBAT on PGA crystallization. Therefore, compounding blends with entangled or network molecular chains contributed to an enhanced interlayer quality for the FDM technique.

In order to further analyze the failure mechanism, the impact fracture surfaces of injection-molded and 3D-printed specimens were observed (Figure 7). It can be seen that ductile fracture occurred in both samples, and the fractured surface was partly concavo-convex. A few pores were observed in the 3D-printed sample surface (marked with a red circle), which was caused by layer-by-layer deposition during the printing process. Previous studies have demonstrated that the mechanical properties of 3D-printed samples could obtain ~80% of the injection-molded counterparts, which was restricted by the left voids or interface between the adjacent printing layers. Therefore, it is acceptable for slightly lower values of mechanical properties for PGA/PBAT (85/15) blend feedstock in comparison with IM materials. Nevertheless, the 3D-printed PGA/PBAT (85/15) filament possesses great potentials in fabricating complex structures.

### 3.3. Applications for PGA/PBAT Structure Manufacturing

It was demonstrated that TPMS structures possessed excellent energy absorption capacity, and the graded-thickness samples could avoid large stress fluctuations and show high cumulative energy absorption values than the constant-thickness samples [41,42]. Among the known TPMS structures (i.e., Diamond (D), Gyroid (G), I-WP, etc.), the stiffness, yield strength, ultimate strength and energy absorption capacity were investigated [43]. Results indicated that the TPMS-D structure exhibited excellent compressive property. Therefore, uniform and graded TPMS structures of D surfaces were adopted in this study. Images of geometric D-TPMS models, 3D-printed samples and the CT-reconstruction were assembled in Figure 8. Uniform pore architecture with two sheet thicknesses and radially graded structures were designed in the current work. The 3D-printed samples were highly coincident with geometric models, suggesting FDM printing was able to fabricate complex D-TPMS structures. Furthermore, 3D-reconstructed CT images obtained the three-dimension structure, confirming the desired structure was achieved by the FDM printing using PGA/PBAT (85/15). All samples possessed a well-controlled 3D porous structure with high interconnectivity. In 2D-reconstructed CT images, the cross-section images in the x-z and x-y planes clearly showed the internal pore architecture. In the constant-thickness D-TPMS structure, the sheet thickness was quite uniform, whereas the thickness from thin to thick was radially distributed in the cylinder.

The stress-strain curves of the printed D-surfaces of neat PGA and PGA/PBAT (85/15) were presented in Figure 9a. For neat PGA, the compressive curve showed a sudden drop when the stain only reached 5% with a corresponding compression strength of 24.2 MPa, indicating an inherent brittleness of PGA. During the compression test, PGA D-surface structures broke into pieces when suffered compression loading force. In contrast, PGA/PBAT (85/15) structures displayed a continuous profile in the stress-strain curves, suggesting an improved toughness of composite materials. The maximum compressive strength was 29.9 MPa for PGA/PBAT (85/15), which was 25% higher than that of PGA. Accordingly, it was determined that PGA/PBAT had better resistance to compression loading force. Although the mechanical strength of neat PGA was stronger than PGA/PBAT blend, the brittleness constrained the applications when used as energy absorbers. The energy absorbed per mass of graded D-TPMS structure was calculated up to compressive strains to 0.2 and plotted in Figure 9b. The cumulative SEA continuously increased with the increase of compressive strain. After the compression test, samples exhibited a mixture failure mode of several delaminations and local cracks occurred at the bottom, which might be caused by minor stress fluctuations.

Figure 10 displayed the FDM printed complicated parts using PGA/PBAT filament feedstock, proving that the PGA/PBAT blend filament can be 3D-printed into geometrically complex parts. In addition, vertically tall and slender cylinders were successfully printed with fine surfaces. Therefore, biodegradable PGA and PBAT polymer were suitable to manufacture high strength, lightweight and complex shaped parts, providing alternative materials of green plastics in the printing materials.

## 4. Conclusions

The research demonstrated that the binary PGA/PBAT (85/15) blend cross-linked by ADR chain extender was of good printability as FDM 3D printing feedstock. Utilizing biodegradable PGA/PBAT blends shows great potential in products or prototypes with a prospect of environmental protection value.

In conclusion:(1)The crystallization process of composite filament was affected by blending of PBAT, and thermal stability of PGA/PBAT (95/5, 85/15) were superior to neat PGA whereas that of PGA/PBAT (75/25) became deteriorated. The utilization of an ADR chain extender can improve the compatibility of PGA and PBAT to some extent.(2)The incorporation of PBAT decreased the tensile strength and modulus but effectively enhanced the elongation at the break of PGA/PBAT blends, achieving an improved toughness. The mechanical properties (including stiffness, toughness) could be well tailored by changing the formulations.(3)3D-printed PGA/PBAT (85/15) were successfully fabricated into filaments, and the mechanical performance of printed samples was close to that of injection-molded counterparts.(4)D-TPMS structures with uniform and graded pore architectures were designed and manufactured. The graded-thickness PGA/PBAT TPMS samples exhibited good stiffness, strength and energy absorption capacities.

Future work could be focused on the development of wider compositions of PGA/PBAT blend filament for 3D printing and investigate the energy absorption between graded structures and non-graded structures.

## Figures and Tables

**Figure 1 polymers-13-03757-f001:**
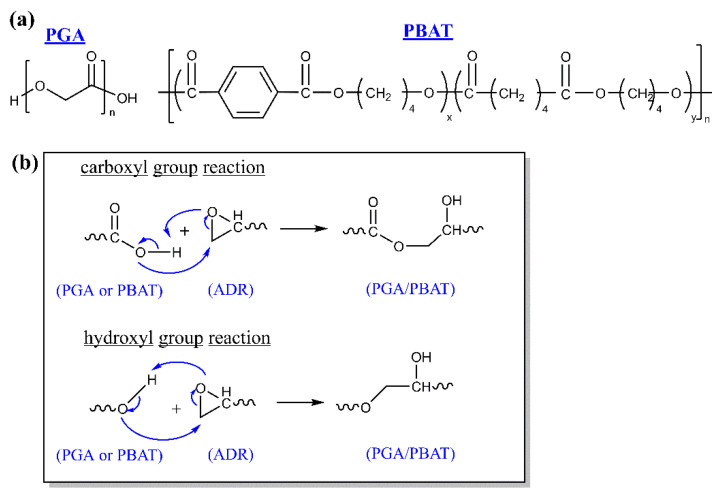
(**a**) Chemical structure of PGA and PBAT, (**b**) Reaction mechanism between PGA and PBAT with ADR.

**Figure 2 polymers-13-03757-f002:**
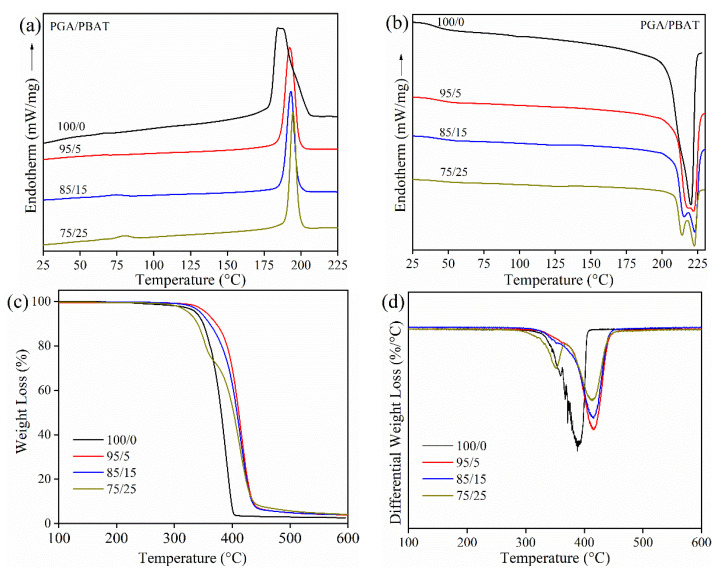
(**a**) Cooling; (**b**) Second heating DSC curves; (**c**) Weight loss; (**d**) Differential thermogravimetric curves of PGA and PGA/PBAT composite pellets.

**Figure 3 polymers-13-03757-f003:**
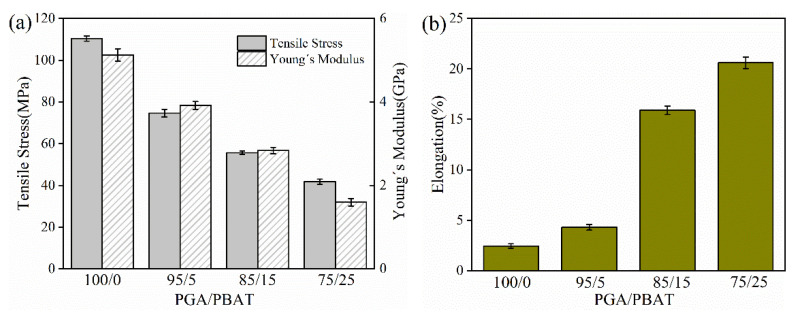
(**a**) Tensile strength, Young’s modulus; (**b**) elongation at break of PGA/PBAT (100/0, 95/5, 85/15 and 75/25) blends.

**Figure 4 polymers-13-03757-f004:**
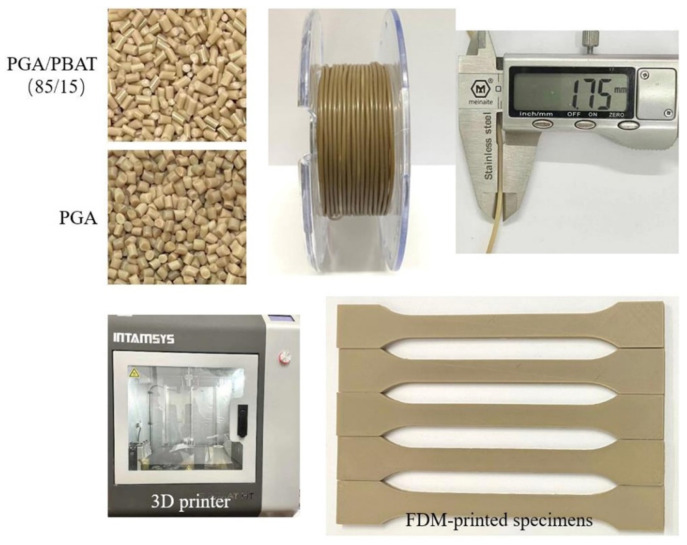
Original PGA and PGA/PBAT (85/15) composite pellets, extruded filaments of PGA/PBAT (85/15) and FDM-printed tensile specimens.

**Figure 5 polymers-13-03757-f005:**
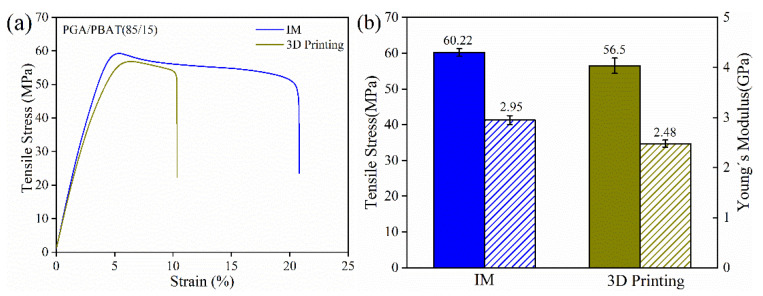
(**a**) Tensile stress-strain curves; (**b**) histograms of tensile strength and Young’s modulus of PGA/PBAT (85/15) specimens fabricated by IM and 3D printing.

**Figure 6 polymers-13-03757-f006:**
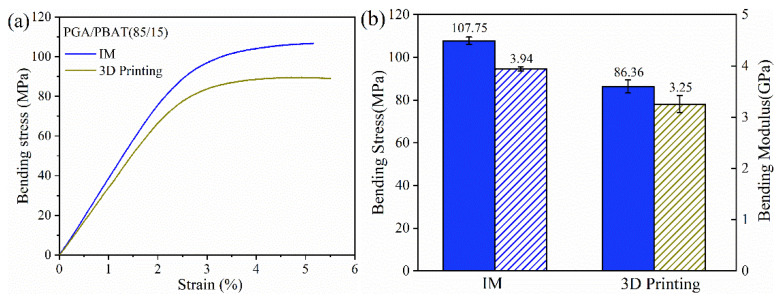
(**a**) Flexural stress-strain curves; (**b**) histograms of bending strength and modulus of PGA/PBAT (85/15) specimens fabricated by IM and 3D printing.

**Figure 7 polymers-13-03757-f007:**
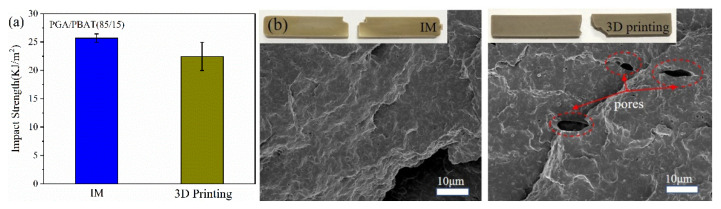
(**a**) Notch impact values; (**b**) fracture surfaces of PGA/PBAT (85/15) specimens fabricated by injection molding and 3D printing.

**Figure 8 polymers-13-03757-f008:**
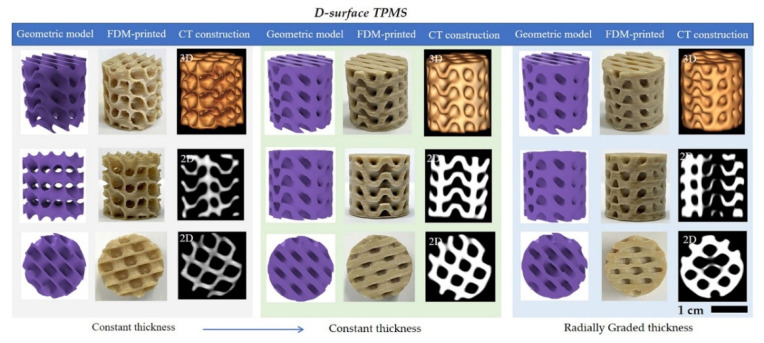
Geometric models, 3D-printed and CT reconstruction images of D-surface TPMS structures with the uniform and radially graded pore architectures.

**Figure 9 polymers-13-03757-f009:**
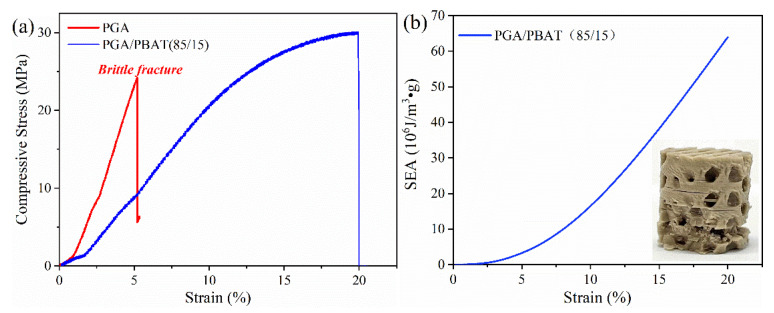
(**a**) Compressive stress-strain curves of PGA and PGA/PBAT (85/15) D-TPMS structures; (**b**) SEA versus strain curves of PGA/PBAT (85/15) structures.

**Figure 10 polymers-13-03757-f010:**
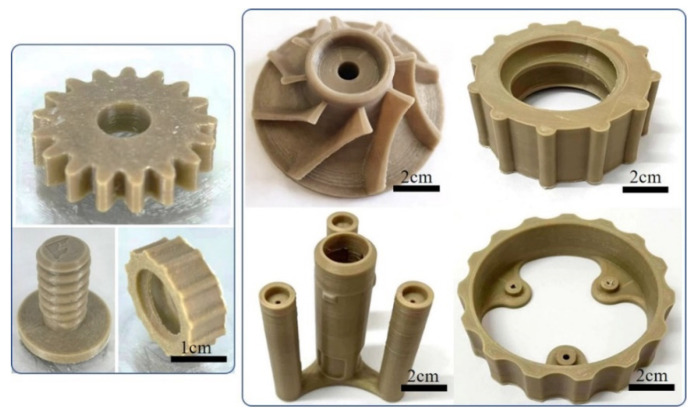
Typical FDM-3D printed PGA/PBAT industrial parts with complex shapes.

**Table 1 polymers-13-03757-t001:** Crystalline and melting and decomposition parameters of PGA, PGA/PBAT (95/5), PGA/PBAT (85/15) and PGA/PBAT (75/25).

Specimen	T*_c_* (°C)	∆H*_c_* (J/g)	T*_m_* (°C)	∆H*_m_* (J/g)	T*_d,5%_* (°C)	T*_d,max_* (°C)
PGA	185.1	76.3	220.5	84.5	339.2	389.7
PGA/PBAT(95/5)	192.0	64.7	217.7/222.2	68.3	358.2	415.7
PGA/PBAT(85/15)	193.3	58.8	215.5/222.3	61.8	346.7	414.4
PGA/PBAT(75/25)	194.5	51.2	214.2/222.4	51.5	326.0	412.5

*T**_d,5%_:* the initial decomposition temperature, 5% of loss weight; *T**_max,5%_:* temperature to the maximum decomposition rate.

## Data Availability

The data presented in this study are available on request from the corresponding author. The data are not publicly available due to on-going relevant study.

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
