# Peer review of "Biodegradable PGA/PBAT Blends for 3D Printing: Material Performance and Periodic Minimal Surface Structures"

_polymers, 2021, doi:10.3390/polym13213757_

Round 1

Reviewer 1 Report

  1. Suggest to label the chemical compounds in Figure 2.
  2. The value of the manuscript can be enhanced if authors provide some insight to improve the limitation of the 3D printing model using layer-by-layer deposition (line 329, 329)
  3. The role of ADR is another important aspect reported in the paper, however, the authors did not discuss in Conclusion. Suggest to mention that in Conclusion.
  4. Please write full words for DTG, explain the detailed procedures if possible. Or, is it presented as the result of TGA? Authors are suggested to clarify in the manuscript.
  5. Suggest to explain the reason for choosing diamond over other TPMS structures to increase the value of the manuscript.
  6. Please check grammar writing for line 62, line 247, line 404, line 346 and caption for Figure 3 (c).

Author Response

We are grateful to the Reviewer for their professional comments and constructive suggestions. After reading through all comments carefully, we have thoroughly revised the manuscript according to the reviewers’ comments. The point-to-point responses to the reviewers’ comments and the changes in the revision are listed in the attachment.

Reviewer 2 Report

The development of 3D printing technology is currently very dynamic.
More and more materials can be used to generate breaches in this way.
This technology is mainly dedicated to prototypes. The authors use mixtures of biodegradable polymers to assess their suitability for 3D printing. It would be worth confirming the biodegradability of the respondents. The authors correctly formulated the problem and the methodology of work. It is not necessary to present photos of the samples themselves for strength tests, the shape complies with the standard, unless the purpose was to show the color of the samples. The tables are legible and the descriptions are in the main text. The conclusions are correctly formulated. The literature review is satisfactory and up-to-date.

Author Response

Author’s reply: Thank you for your approval. As you pointed out, we also focused the biodegradability of the PGA/PBAT blends. Currently, we have prepared PGA/PBAT/HA(hydroxyapatite) filament and printed into D-TPMS. The in-vivo biodegradability and biocompatibility of PGA/PBAT/HA will be especially focused on, and we hope that the biodegradability of the printed samples would meet the requirement of bone scaffolds. Also, we have deleted the present photos of samples in the Figure 5 and Figure 6.